# The utility of borescope and ATP Biofluorescence for inspection of ophthalmic phaco handpiece lumen: a single-center observational study

Meng Zhan[1☯], Zhuoya Yao ᴵᴰ[1,2*☯], Junhui Geng[1‡], Manchun Li[1‡], Lina Ding[1‡]

1 Central Sterile Supply Department, Henan Provincial People's Hospital, Zhengzhou University People's Hospital, Zhengzhou, China, 2 Henan Provincial Medical Key Laboratory for Quality Control of Medical Devices Sterilization, Zhengzhou, China

☯ These authors contributed equally to this work.
‡ These authors also contributed equally to this work.
* zm15093137681@sina.com

## Abstract

The presence of residual debris in the phaco handpiece lumen is implicated in the onset of intraocular inflammation. While ATP bioluminescence testing is accepted for evaluating the cleanliness of surgical instruments, the borescope's ability to facilitate a visual inspection of the internal integrity of phaco handpiece lumens remains underreported. This study aims to evaluate the efficacy of current cleaning protocols and explore the borescope's utility in visually inspecting phaco handpieces. In this analysis, 41 phaco handpieces underwent borescope examination and ATP bioluminescence testing following thorough cleaning, with two inspections completed on each handpiece. Borescope inspections revealed that 56.10% of phaco handpieces harbored foreign materials or exhibited various forms of structural damage, such as corrosion, rust, green lint, and discoloration. However, ATP bioluminescence testing deemed the cleanliness of all handpieces satisfactory. This study demonstrates the value of a borescope in the visual inspection of phaco handpieces, revealing issues such as corrosion and rust that necessitated immediate flushing of the lumen with sterile distilled or deionized water. The detection of green lint led to the prohibition of placing textile fiber-containing sterile textile wipes on the operating table surface by sterile supply staff, enhancing patient safety measures. Future efforts will focus on addressing the challenges posed by corrosion and particulate formation during rinsing.

## Introduction

The prevalence of ophthalmic surgeries has notably increased following advancements in diagnostic and therapeutic ophthalmology. Cataract surgery, a major ocular procedure, utilizes the smallest intraocular surgical instruments. While these

**Data availability statement:** All relevant data are available from the Figshare Database (URL:https://doi.org/10.6084/m9.figshare.28466747.v1)

**Funding:** The author(s) received no specific funding for this work.

**Competing interests:** The authors have declared that no competing interests exist.

instruments are typically minimally contaminated by tissues or bacteria [1], even trace amounts of residual detergents or chemical pollutants, otherwise tolerable in other body sites, can provoke severe consequences within the eye, leading to toxic anterior segment syndrome (TASS) [2,3].

TASS, a non-infectious postoperative inflammatory condition, is commonly observed following cataract surgery but can also develop following corneal transplantation and posterior segment operations [4–7]. Recent studies have identified inadequate flushing, enzymatic detergent usage, metals, viscoelastic contamination, and adverse drug reactions as the primary contributors to TASS development [8–10]. Comprehensive clinical and laboratory examinations often fail to identify pathogens in TASS cases [1,11], highlighting the critical role of thorough instrument cleaning and sterilization practices in preventing TASS. The Central Sterile Supply Department (CSSD) in China oversees the maintenance of all reusable surgical instruments, including ophthalmic instruments. Its responsibilities encompass recovery, sorting, cleaning, disinfection, and sterilization of these instruments to ensure their sterility for reuse [12].

The ophthalmic phaco handpiece, a key tool in cataract surgery, emulsifies the lens nucleus into a form that can be aspirated. If the residual lens cortex and viscoelastic materials are inadequately cleaned from the phaco handpiece, they may transform into toxic agents during sterilization, leading to intraocular toxicity [13]. Guidelines for the Cleaning and Sterilization of Intraocular Surgical Instruments state that instruments must be visually inspected for debris and damage after cleaning and before packaging for sterilization to ensure that debris removal is complete [1]. However, many modern surgical instruments feature complex designs with crevices, hinges, and narrow lumens that hinder the assessment of cleanliness through visual inspection alone [14]. The ophthalmic phaco handpiece presents a similar challenge due to its small lumen. Since the Adenosine Triphosphate (ATP) is found in all living tissues, its concentration serves as an indicator of bioburden, the greater the ATP level, the greater the contamination. ATP bioluminescence testing quantifies fluorescence intensity to estimate ATP levels [15], offering a sensitive and widely employed method for determining cleanliness in CSSDs. In recent years, borescopes have become increasingly utilized in the visual inspection of flexible endoscopes [16,17]. These small, light-equipped cameras allow CSSD staff to examine areas that are otherwise hidden from view, thereby providing an approach for directly assessing the internal cleanliness of lumened instruments [18]. While the utility of borescopes in inspecting endoscopes is well-documented, highlighting their importance for identifying damage, abnormalities, and residual materials [19,20], evidence of their use in phaco handpiece inspections remains sparse.

To enhance the cleaning efficacy of phaco handpieces, our CSSD utilized both a borescope and ATP bioluminescence for lumen assessments. The study primarily aimed to (1) evaluate the borescope's effectiveness in facilitating the visual inspection of phaco handpieces and (2) scrutinize the CSSD's existing reprocessing protocols to determine opportunities for ongoing process improvement.

## Methods

### Settings

The study was conducted in a large tertiary general hospital and its associated eye hospital, which houses 12 ocular surgery suites with an annual surgical volume of over 20,000 cases, has 184 open beds, and sees an annual outpatient volume of nearly 500,000. In this setting, all used ophthalmic surgical instruments, including phaco handpieces, are centrally processed in the CSSD. This study was conducted from May to August 2023 at our CSSD. The project was approved by the medical ethics committee of Henan Provincial People's Hospital(Ethical Clearance No. 125 of 2023).

### Phaco handpieces

Titanium alloy phaco handpieces (model BL3170, Bausch & Lomb) were utilized in our institution. They had two lumens: one for irrigation and the other for aspiration. The cleaning of phaco handpieces was conducted manually. Initially, these instruments were pretreated by nurses in the ophthalmic operating room before being transported to the CSSD in a sealed transfer box. Within the designated ophthalmic processing area of the CSSD, staff followed the manufacturer's recommendations and the Ophthalmic Surgical Instruments Cleaning, Disinfection, and Sterilization Technology Operation Guide [21] to carry out the cleaning process. This process involved rinsing the surface of the instrument with running water, followed by the insertion and retraction of a 50 ml syringe into both the irrigation and aspiration lumens of the phaco handpiece to ensure thorough internal cleaning. Subsequently, the handpieces were immersed in a neutral pH detergent (YOMA) solution, with particular attention given to scrubbing the lumens and threads with a brush beneath the liquid surface to remove any adherent residues. Following this cleaning, a comprehensive rinse was performed, culminating in a final rinse to ensure the elimination of all residual substances. The subsequent phase involved both lumens being subjected to air-drying by utilizing jet guns, each for a duration of 30 seconds, to remove any residual moisture. The cleanliness of the handpieces was then evaluated through both borescope inspections and ATP bioluminescence testing to ascertain the efficacy of the cleaning process.

### Borescope inspection

A borescope is a visual inspection system that utilizes a thin, flexible wire equipped with a camera and light source to detect residual materials and contamination within luminal instruments [22]. The apparatus (Shenzhen Xinruida) comprised a tablet PC running proprietary software and a borescope with an operating handle. A data cable was used to connect the two components. A 1.2-meter-long, 1.8-millimeter-diameter, high-precision borescope with a lens resolution of up to 300,000 pixels was inserted into the instrument's lumen by the examiner, who also used the handle to take pictures and videos and the tablet to visualize them. In this project, borescope inspections were systematically carried out by the CSSD staff, with the examination frequency based on the daily procedural workload, ensuring a broad temporal spread for accuracy in results. The examiner first inserted the borescope into the inlet of the irrigation lumen until reaching the end of the lumen. Thereafter, insertion into the aspiration lumen was performed. Because of the inner diameter of the aspiration lumen, only the inlet and outlet regions were inspected. Each handpiece inspection lasted approximately 3 minutes, with the borescope sanitized using 75% ethanol between uses (Fig 1). Notably damaged handpieces were identified for subsequent repair.

### ATP Biofluorescence

The Association for the Advancement of Medical Instrumentation (AAMI) recommends that instruments be verified by a cleaning compliance test after completion of the cleaning procedure [23]. ATP biofluorescence is a fast and reliable test for the quantitative assessment of cleaning quality. The 3M Clean Trace NGi ATP biofluorescence detector was utilized in this study, which exhibited high repeatability and sensitivity, and had been validated by an independent third-party body in the UK. The standard operating procedure employing a 3M™ Clean-Trace™ liquid sampling swab was followed. During the process, the phaco handpiece was held to facilitate the filling of both the irrigation and aspiration tube lumens

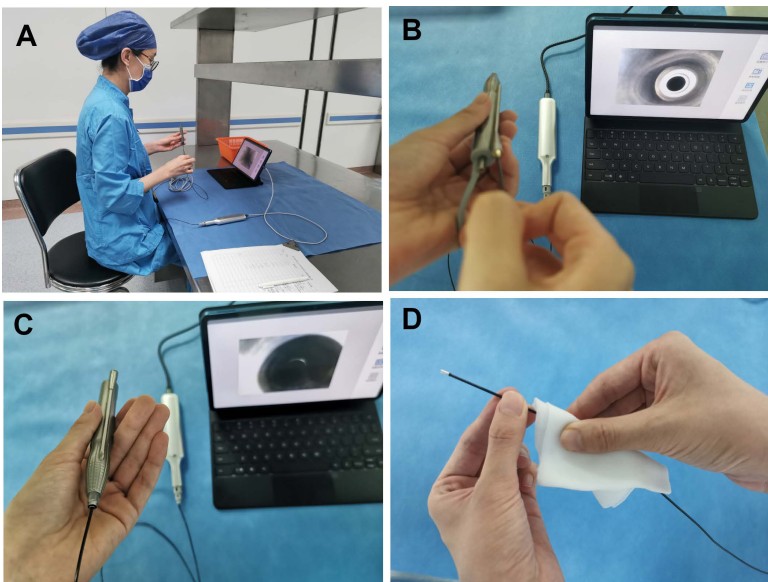

**Fig 1. Borescope inspection procedure.** A)Connect the apparatus correctly. B)Inspect the irrigation lumen. C)Inspect the aspiration lumen. D)Wipe and disinfect the borescope with 75% ethanol.

with approximately 10 ml of sterile water each by CSSD staff. This sterile water was then collected into a sterile container under the phaco handpiece. Care was taken to moisten the swab head with the rinse solution collected in the sterile container without touching the inner walls of the sterile container. Once moistened, the swab was transferred to a test tube, agitated, and then introduced into the ATP biofluorescence detector for analysis (Fig 2). According to the testing protocol, a reading of ≤ 150 Relative Light Units (RLU) indicated satisfactory cleaning [24].

## Statistical analyses

Statistical analyses were performed using IBM SPSS Statistics software (version 26.0). Normally distributed continuous variables are reported as *Mean ± SD* and compared across multiple groups using *ANOVA.* Categorical variables are reported as frequencies with percentages, with between-group differences assessed using *Chi-square tests.* A significance threshold of *P < 0.05* was set to determine statistical significance.

## Results

### Phaco handpieces characteristics and overall findings

In total, 41 phaco handpieces were subjected to both examinations, with a service life of 2–8 years and a mean value of $5.39 \pm 1.61$ years. Eighteen cases passed all inspections, 17 cases had one finding in the borescope's visual inspection, and six handles had multiple abnormal findings (Table 1). The difference in the number of years of usage between the handles was statistically significant ($F = 7.384$, $p < 0.05$).

### Borescopes inspection

In 23 of the cases (56.10%), the borescope inspections of phaco handpieces revealed significant findings within the lumens, such as corrosion, rust, green lint, and discoloration (Table 2). Up to 53.66% of the irrigation lumens of the phaco handpieces were problematic, suggesting a significant difference compared to the aspiration lumens ($\chi^2 = 12.004$, $p < 0.05$).

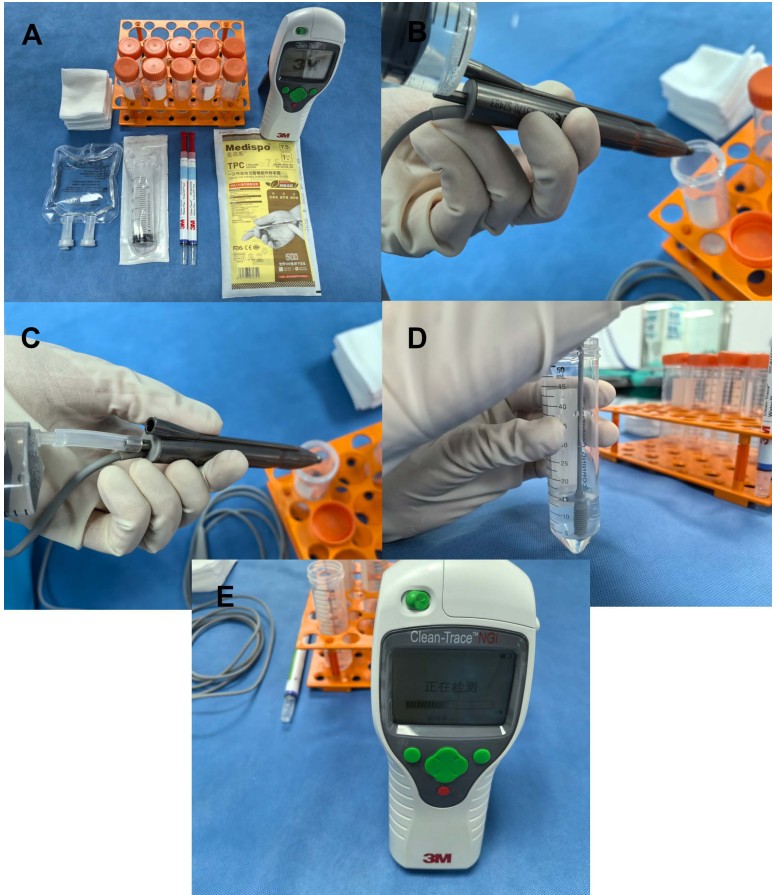

**Fig 2. Procedure for ATP biofluorescence test using a liquid sampling swab.** A)Prepare the items. B)Flush the irrigation lumen.C)Flush the aspiration lumen.D)Sample the rinse solution with a liquid swab.E)Read the data.

**Table 1. Comparison of the number of significant findings for phaco handpieces*.**

| Groups | Cases | Age (years) | F | P-value |
|---|---|---|---|---|
| No finding | 18 | 4.44 ± 1.69 | 7.384 | 0.002 |
| One finding | 17 | 6.06 ± 1.03 | | |
| Multiple findings | 6 | 6.33 ± 1.37 | | |
| Total | 41 | 5.39 ± 1.61 | | |

*As all phaco handpieces passed the ATP biofluorescence assay, the numbers in the table are the result of visual inspection by borescopes.

**Findings within the irrigation lumens.** Pitting corrosion of varying degrees was observed in 10 cases (24.39%) within the irrigation lumen, predominantly at the interface, mid-lumen, and outlet, appearing as flaky formations (Fig 3). Rust corrosion was detected in 7 cases (17.07%), mainly at the articulation threads (Fig 4). Discoloration along the lumen's interior wall was noted in 5 cases (12.20%) (Fig 5).

**Findings within the aspiration lumens.** Rust was identified at the interface and articulation of the aspiration lumen in 5 cases (12.20%). Notably, green lint was discovered within the aspiration lumen in 2 instances (4.88%) (Fig 6),

**Table 2. Comparison of findings among different lumens of phaco handpieces*.**

| Site | Finding Items*(%) | | | | Total | χ2 | P-value |
|---|---|---|---|---|---|---|---|
| | Corrosion | Rust | Green lint | Discolouration | | | |
| Irrigation lumen (n=41) | 10(24.39%) | 7(17.07%) | – | 5(12.20%) | 22(53.66%) | 12.004 | 0.001 |
| Aspiration lumen (n=41) | – | 5(12.20%) | 2 (4.88%) | – | 7 (17.08%) | | |

*These results represent objective findings only and are not indicative of severity.

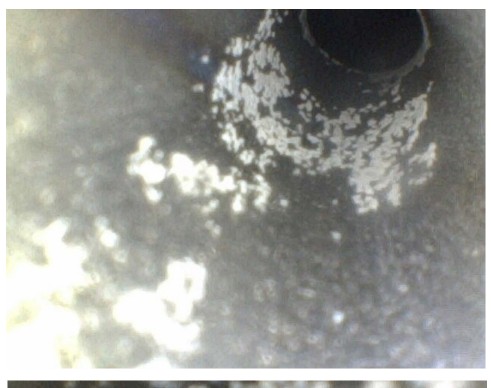

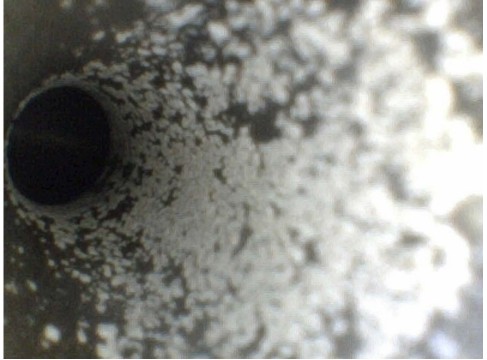

**Fig 3. Pitting corrosion within the irrigation lumen.**

likely originating from sterile wipes made of textile material used on the operating table, which could have entered the lumen during air drying. This finding led to the discontinuation of textile sterile wipe use on the operating table to prevent potential ocular washout risks.

## ATP Biofluorescence

The ATP biofluorescence tests for all 41 phaco handpieces showed RLU values below 150, indicating satisfactory cleaning.

## Discussion

The phaco handpiece, a critical tool in ophthalmic surgeries such as cataract removal, demands rigorous cleaning procedures due to its complex internal structure. Our CSSD emphasizes the careful cleaning of these handpieces, following

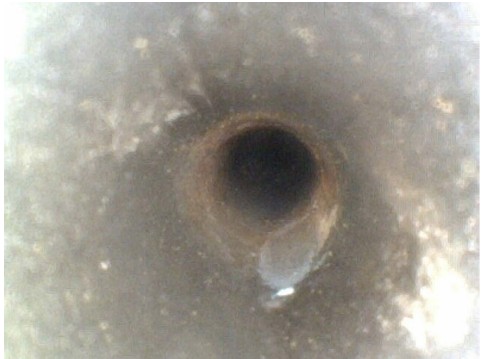

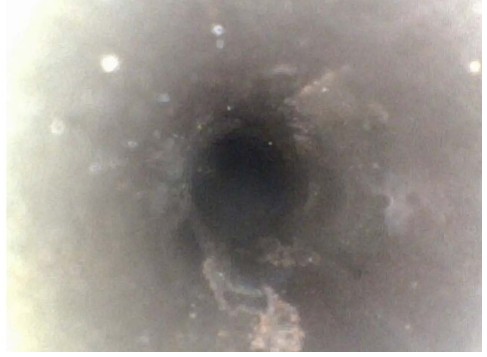

**Fig 4. Rust corrosion within the irrigation lumen.**

specific protocols that include isolated cleaning processes, designated processing areas and using suitable detergents to minimize risks arising from cross-contamination and residue.

Traditionally, handpiece inspections have relied on external examinations utilizing light-ray magnifying glasses, which are less effective for assessing the narrow internal lumen. As a result, quantitative methods such as ATP bioluminescence and residual protein testing have become standard for evaluating lumen cleanliness [23]. In this study, we incorporated ATP bioluminescence testing alongside borescope examinations, providing a detailed view of the internal condition of the handpieces, which helped bridge the gap between superficial and thorough internal assessments.

The study identified pitting corrosion in 24.39% of the handpieces, potentially related to the balanced salt solutions used for irrigation during surgery; these contain electrolytes that could exacerbate corrosion over time [18]. Furthermore, 12.20% of the handpieces exhibited rust, particularly at the entry and articulation points, raising concerns about bacterial adhesion and biofilm formation on the instrument surfaces. Instrument corrosion and rust production are slow processes, and "particle" formation can be seen in the initial stages of this process [25]. Many scholars who have identified metallic fragments in the eye have reported that the main component of the metal fragments is titanium, which is consistent with the composition of the handpieces and tips [26–28]. Martinez-Toldos et al. [27] found that the vibration of the phaco hand-piece, which is subjected to frequent use, can lead to the dislodgement of these metal particles into the eye due to wear and tear on the handles. In previous cases, metal fragments were well tolerated in the eye, but a case report described recalcitrant postoperative inflammation due to metal fragments on the surface of the iris [29]. Paolo et al. [28] suggested that even if sterile, the fragments can potentially trigger inflammation and lead to TASS. However, there is controversy regarding the origin and hazardous effects of these metal particles. In response to these findings, ophthalmic operating

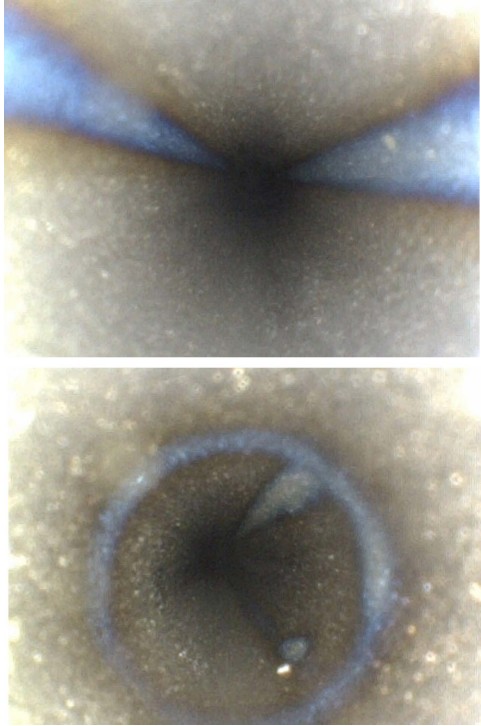

**Fig 5. Discoloration within the irrigation lumen.**

theater nurses were advised to rinse the handpieces promptly with sterile distilled or deionized water post-surgery to minimize electrolyte residue. Additionally, the discovery of green lint in the aspiration lumen prompted a change in drying practices, specifically prohibiting the use of sterile textile wipes on the operating table to avoid lint contamination in the handpieces.

Furthermore, the topic of instrument aging is broad. Poor functioning is associated with instrument aging. However, corrosion, rust, and discoloration did not impair the applicability of the instruments in this study. Malfunctioning ultrasound transducers, connector wires, and connectors are the primary causes of instrument repair [30]. As shown in Table 1, initial lumen-related damage was observed at a mean service duration of 6.06 ± 1.03 years (Mean ± SD). Based on these findings, we recommend routine inspections for handpieces exceeding five years of clinical use, with a particular focus on irrigation lumen integrity. This proactive maintenance approach allows for the early detection of structural deterioration, ensuring patient operational safety for patients.

ATP bioluminescence testing yielded satisfactory results, confirming the efficacy of our standardized decontamination protocols, which align with the manufacturer specifications and evidence-based guidelines from ophthalmic professional associations. However, as ATP bioluminescence detects residual organic contamination detection, its sensitivity is limited when applied to phaco handpieces where bioburden levels remain low. Furthermore, ATP bioluminescence is less effective than borescope inspection for detecting structural damage. Routine borescope examinations revealed persistent structural defects that, despite meeting cleanliness standards, could serve as potential biofilm reservoirs [25]. These findings underscore the critical need for a multimodal quality assurance strategy. We propose implementing a comprehensive surveillance system that monitors microbial contamination and device integrity.

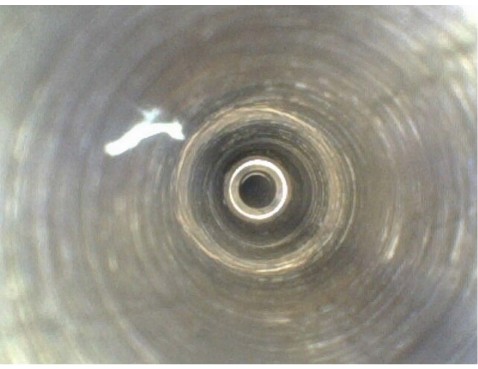

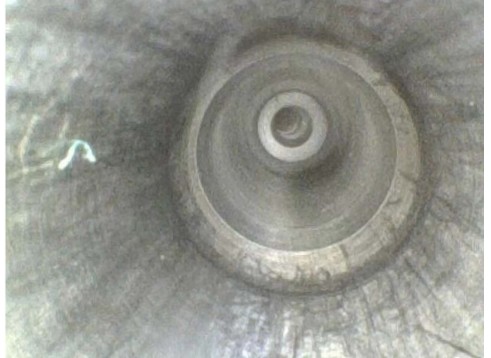

**Fig 6. Green lint within the aspiration lumen.**

## Strengths and limitations

This study demonstrates previously unrecognized cleaning challenges and highlights three key preventive measures: 1) prompt post-surgery irrigation of instrument channels with sterile distilled water, 2) implementation of controlled-environment protocols for compressed-air drying to minimize particulate contamination, and 3) establishment of scheduled maintenance cycles to preserve lumen integrity. However, our study has some limitations. First, the clinical significance of these structural abnormalities remains unclear. Our hospital's assessments after cataract surgery did not show any cases of intraocular metallic fragments. Nevertheless, these results should serve as a helpful reminder of potential dangers. Second, this was a single-center study, which should be expanded in the future to incorporate data from a larger sample size.

## Conclusion

This study highlights the value of borescope inspection as part of a visual inspection of phaco handpiece integrity, identifying issues such as corrosion and rust that necessitated immediate lumen flushing with sterile distilled or deionized water during surgery. The detection of green lint prompted the discontinuation of textile fiber-containing wipes on the operating table, thereby enhancing patient safety measures. Future advances are expected to integrate both qualitative and quantitative approaches for a more comprehensive evaluation of instrument cleanliness. Additionally, ongoing efforts will focus on mitigating corrosion and particulate formation during rinsing to further enhance the effectiveness of cleaning protocols.

## Acknowledgments

We wish to thank the nurses in the Ophthalmic Operating Room of Henan Provincial Eye Hospital for supporting this study. The authors also thank the CSSD staff of Henan Provincial People's Hospital for their cooperation and support.

## Author contributions

**Conceptualization:** Meng Zhan, Zhuoya YAO, Junhui Geng.

**Data curation:** Meng Zhan, Junhui Geng.

**Formal analysis:** Meng Zhan, Zhuoya YAO, Lina Ding.

**Funding acquisition:** Junhui Geng, Lina Ding.

**Investigation:** Lina Ding.

**Methodology:** Zhuoya YAO.

**Supervision:** Manchun Li.

**Validation:** Zhuoya YAO, Manchun Li.

**Writing – original draft:** Meng Zhan, Zhuoya YAO.

**Writing – review & editing:** Meng Zhan, Zhuoya YAO, Junhui Geng, Manchun Li, Lina Ding.

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
