## [Decision Letter · Decision Letter 0]

22 Jan 2025

PONE-D-24-50911The utility of borescope and ATP Biofluorescence for inspection of ophthalmic phaco handpiece lumen: a single-center observational studyPLOS ONE

Dear Dr. YAO,

Thank you for submitting your manuscript to PLOS ONE. After careful consideration, we feel that it has merit but does not fully meet PLOS ONE’s publication criteria as it currently stands. Therefore, we invite you to submit a revised version of the manuscript that addresses the points raised during the review process.

We look forward to receiving your revised manuscript.

Kind regards,

Yik-Ling Chew

Academic Editor

PLOS ONE

Journal Requirements:

3. Please upload a copy of Supporting Information Figure/Table/etc. S1 Fig.1 and S2 Dataset which you refer to in your text on page 11.

Reviewers' comments:

Reviewer's Responses to Questions

**Comments to the Author**

1. Is the manuscript technically sound, and do the data support the conclusions?

Reviewer #1: Partly

2. Has the statistical analysis been performed appropriately and rigorously? 

Reviewer #1: No

3. Have the authors made all data underlying the findings in their manuscript fully available?

Reviewer #1: Yes

4. Is the manuscript presented in an intelligible fashion and written in standard English?

Reviewer #1: No

5. Review Comments to the Author

Reviewer #1: The concept of the research was good but it could be in more presentable format.

6. PLOS authors have the option to publish the peer review history of their article (what does this mean? ). If published, this will include your full peer review and any attached files.

**Do you want your identity to be public for this peer review?** For information about this choice, including consent withdrawal, please see our Privacy Policy .

Reviewer #1: **Yes: ** Sifat Uz Zaman

---

## [Author Response · Author response to Decision Letter 0]

5 Mar 2025

Dear Editors and Reviewers,

We appreciate the opportunity to revise our manuscript titled "The utility of borescope and ATP Biofluorescence for inspection of ophthalmic phaco handpiece lumen: a single-center observational study" and are grateful for the insightful comments provided by the reviewers. Those comments are all valuable and helpful for revising and improving our paper, as well as the important guiding significance to our research. In the following, we have provided detailed responses to each reviewer's comments. Revised portions are marked in red on the paper. Additionally, we have conducted a comprehensive revision of the entire manuscript. In this response letter, the reviewers' comments are presented in italics, and our corresponding changes and additions to the manuscript are highlighted in red text. We have tried our best to make all the revisions clear, and we hope that the revised manuscript meets the requirements for publication.

---

## [Editor Report · Decision Letter 1]

6 Apr 2025

The utility of borescope and ATP Biofluorescence for inspection of ophthalmic phaco handpiece lumen: a single-center observational study

PONE-D-24-50911R1

Dear Dr. Zhuoya Yao,

We’re pleased to inform you that your manuscript has been judged scientifically suitable for publication and will be formally accepted for publication once it meets all outstanding technical requirements.

Kind regards,

Yik-Ling Chew

Academic Editor

PLOS ONE
---

## [Editor Report · Acceptance letter]

PONE-D-24-50911R1

PLOS ONE

Dear Dr. YAO,

I'm pleased to inform you that your manuscript has been deemed suitable for publication in PLOS ONE. Congratulations! Your manuscript is now being handed over to our production team.

Kind regards,

on behalf of

Dr. Yik-Ling Chew

Academic Editor

PLOS ONE